# Multilevel Modeling of Individual and Group Level Influences on Critical Thinking and Clinical Decision-Making Skills among Registered Nurses: A Study Protocol

**DOI:** 10.3390/healthcare11081169

**Published:** 2023-04-19

**Authors:** Nur Hidayah Zainal, Kamarul Imran Musa, Nur Syahmina Rasudin, Zakira Mamat

**Affiliations:** 1Department of Community Medicine, School of Medical Sciences, Health Campus, Universiti Sains Malaysia, Kubang Kerian 16150, Kelantan, Malaysia; 2Biomedicine Programme, School of Health Sciences, Health Campus, Universiti Sains Malaysia, Kubang Kerian 16150, Kelantan, Malaysia; 3Nursing Programme, School of Health Sciences, Health Campus, Universiti Sains Malaysia, Kubang Kerian 16150, Kelantan, Malaysia

**Keywords:** critical thinking skill, clinical decision-making skill, registered nurse, multilevel modeling

## Abstract

Critical thinking and clinical decision-making are two essential components of nursing services. The practice of nursing requires both components at every level of nurses’ daily tasks. This paper describes the protocol for an ongoing project, which aims to examine the prevalence of critical thinking and clinical decision-making among registered nurses and determine the factors that influence the skills at individual and group levels using a multilevel modeling approach. Survey data will be collected from approximately nine states, with nine general hospitals, nine district hospitals, one private hospital, and one educational hospital in Malaysia included. We aim to recruit 800 registered nurses working shifts in hospital settings. We will use questionnaires to measure the perceived nurses’ knowledge, critical thinking skills, and clinical decision-making skills. The study will have three levels as the nurses are nested within the unit and further nested within the hospitals. This study will give an insight into the nursing profession today where critical thinking and clinical decision-making skills can play a significant role in patient safety and the quality of care in nursing practice.

## 1. Introduction

Critical thinking (CT) and clinical decision-making (CDM) are two essential components of nursing services. The practice of nursing requires both components at every level of nurses’ daily tasks. It was depicted by Florence Nightingale in the 19th century [1], recognized by the World Health Organization (WHO), and included in the State of the World’s Nursing 2020 Report. The report emphasized the critical thinking skills in nurses that can contribute to more advanced judgment in clinical decision-making and increase the safety of care provision [2,3].

### Background Research

The requirement to assess CT and CDM skills among nurses and measure the effects of such skills is widely debated [4,5,6]. Research in this area is active because CT and CDM skills are crucial for nurses. The skills are necessary for nurses to develop quality plans of care for patients, to effectively cope with advancing technologies, human resource limitations, the high level of acuity required in diverse healthcare settings, and most importantly, to improve the care outcomes of patients [7,8,9]. Concerning competencies related to CT and CDM skills in nursing, there are calls for researchers to study competency profiles and relevant frameworks and the factors or variables that determine both skills and the quality of nursing services [10,11].

Performance evaluation and measurement of skills are crucial. Therefore, many recent measurement models have been developed to study these skills. Nestel [12] used a multilevel evaluation strategy model to elicit development in trainees’ knowledge, attitudes, and skills and detected sustained changes in clinical practice. This study showed positive directional changes in technical, teaching, and communication skills. Meanwhile, Meghdad [13] used a multi-criteria decision-making model to assess the performance of nurses. It reported that human skills, identification of strengths and weaknesses, suitability of patient relationships, and partnership with colleagues earned the top score. These models have focused on the sociodemographic or individual level as the predictive factor (individual-factor level). Instead of focusing only on individual-level factors, this present study will apply a multi-level modeling framework that focuses on individual- and group-level predictive factors. Multilevel models recognize the existence of such data hierarchies by allowing for residual components at each level in the hierarchy. For example, a three-level model that allows for grouping nurses’ clinical decision-making outcomes within hospitals would include residuals at the nurse, unit, and hospital levels.

Most of the literature has focused on CT and CDM levels and factors from the individual level that influence the skills [14,15]. Given the evidence that both individual and organizational characteristics play significant roles in patient safety and quality of care, it is essential to not only include measures of both individual-level and group-level factors using the multilevel approach. Hence, in this paper, we describe the protocol for a multilevel modeling framework, including the review and field study phases, to assess the most significant factors at the individual and group levels that may influence CT and CDM skills among registered nurses. This study aims to assess CT and CDM skills among registered nurses in Malaysia. In the phase one of the study, we will obtain a comprehensive review of CT and CDM skills using a systematic review. For the phase two, there are four objectives, (a) to determine the prevalence of CT and CDM skills among registered nurses in Malaysia and (b) to identify the significant predictors for CT and CDM at the individual and group levels among nurses in Malaysia, (c) to assess the potential effect of the potential mediator and moderator on the skills and (d) to compare CT and CDM performances between three different hospital settings.

## 2. Materials and Methods

### 2.1. Study Setting and Population

This study will be conducted in six regions throughout Malaysia: northern, central, southern, the east coast, Sabah, and Sarawak. The total number of registered nurses in Malaysia is around 108,000 [16]. This study will recruit registered nurses from grades U29 and U32, working shifts in hospital settings and having a minimum diploma as the basic qualification. We will exclude nurses who work office hours and are involved in administrative work, in-service education units, infection control units, central sterile supply units, and clinics.

### 2.2. Sample Size Estimation

The minimum sample size was calculated using the Cochran formula with around 384 minimum samples [17]. Additional sample size using the power analysis for multilevel modeling study were determined using **Simr** and **lme4** in R packages [18]. With for the significance level (alpha) set at 0.05, the sample size will achieve 94% power for the model (between hospitals) and 89% power for the model (within hospitals) at 95% CI. However, to ensure that the sample becomes more representative, to cover for possible missing data and to perform multiple comparisons during analysis, an additional 10% from the calculated sample size has been added. Hence, the total number of registered nurses to be recruited will be 800 (20 nurses from each unit).

### 2.3. Sampling Method and Subject Recruitment

The source population is registered nurses from 20 hospitals. The participants will be recruited using multistage stratified cluster random sampling [19]. First, we will obtain the list of government hospitals from the official portal of the Ministry of Health. Next, we will list and stratify the hospitals based on six regions (northern, central, southern, east coast, Sabah, and Sarawak). We will then classify each region as a cluster and select hospitals by region size for general hospitals (GH) and randomly for district hospitals (DH) by using simple random sampling. However, we will use purposive sampling for the teaching and private hospitals due to the small number of teaching and private hospitals available. There will be nine general hospitals, nine district hospitals, one teaching hospital, and one private hospital. In every hospital, we will select two departments or units: medical and emergency. Next, we will randomly select participants from the nurses’ lists. Figure 1 and Figure 2 depict the sampling method for this study.

### 2.4. Data Collection Method

Each participant will be given explicit information about the study background, the purpose of the study, the risks and benefits, and the data collection process. Informed consent will then be obtained before they can enrol on the study. The recruitment e-poster and the online data collection will be distributed online using Google Forms. To protect the confidentiality, participants will not be required to sign into an account to complete the survey. Participation in this study is voluntary. Those not willing to participate will be excluded from the study. The benefit of online form is that the results can be collected easily, rapidly, and inexpensively. The form contains the study's objectives, instructions and data about the demographic profile. The online form is expected to be completed within 10–20 min. The participants’ identification information, such as their name and identity card number will not be recorded. The data collection period will be ongoing for 12 months from the date the link is distributed to each hospital. At the end of the 12 months, the link will be disabled, and no other person except the researchers will be able to access it. 

### 2.5. Study Flow Chart

In phase 1 we will conduct a systematic review to assess CT and CDM skills among registered nurses. The review will assess the level of both skills using the international and local studies, assess the current predictive variables, and look for the study designs and instrumentation used in previous studies. In phase 2, we will perform a cross-sectional study to determine the prevalence of CT and CDM skills and to identify the predictive factors (that may influence both skills). The flowchart of the study is shown in Figure 3.

### 2.6. Conceptual Framework

Generally, nurses are a hierarchical structure nested within a unit or department and further nested within hospitals; the responses obtained from each unit within the same cluster usually bear some similarity due to the influence of a common context; thus, statistical dependencies may occur [20]. Hence, this study will address this dependency by examining registered nurses’ critical thinking and clinical decision-making skills using a multilevel modeling approach. The multilevel analysis begins with the determination of randomly varying outcome parameters which includes a variation in the level of the outcome (intercepts). Next, the analysis will determine within-group relationships which are indicated by the regression coefficients (slope) across the groups [21]. This study will identify the variation of a nurse’s clinical decision-making score as randomly varying outcome parameters. In Figure 4, the framework of multilevel modeling of critical thinking and clinical decision-making skills is shown.

H1: The nurses’ knowledge significantly influences their critical thinking score;H2: Critical thinking significantly influences the clinical decision-making score;H3: Critical thinking mediates the relationship between the nurses’ knowledge and clinical decision-making;H4: In-service training moderates the relationship between the nurses’ knowledge and critical thinking skills;H5: In-service training moderates the relationship between critical thinking and clinical decision-making skills;H6: The group level in level 3 (hospital settings and hospital size) has a significant influence on CDM;H7: The group level in level 2 (unit or department) has a significant influence on CDM;H8: The individual-level factors in level 1 (sociodemographic) have a significant influence on CDM.

### 2.7. Instrument

The self-administered questionnaire contains four sections: (a) demographic data, (b) nurses’ perceived knowledge, (c) critical thinking skills, and (d) clinical decision-making skills. The nurses’ perceived knowledge instrument will be used to measure the nurses’ perceived knowledge regarding CDM skills [22]. This questionnaire was developed based on the situated clinical decision-making framework by Mary Gillespie. It consists of 22 items. We have chosen this instrument because it focuses on measuring nurses’ perceived knowledge regarding their CDM skills. Next, a validation study will be conducted to provide validated and reliable instruments for local respondents. Nur Hidayah Zainal developed the critical and clinical decision-making scale (CTCDMS) based on the 4-Circle Critical Thinking Model and the conflict theory model of decision-making [23]. We chose this scale because it explicitly measures the CT and CDM of nurses in clinical practice. The critical thinking construct consists of 11 items with two factors: critical characteristic (4 items) and critical knowledge (7 items).

In contrast, the clinical decision-making construct consists of 10 items with two factors: decision abilities (5 items) and decision accuracy (5 items). There are interval scales of five units reflecting the subject's perceived quality. Scale one is for strongly disagree, while scale 5 for strongly agree. We will sum the scores and then group them into 3 categories, (a) low (11–25), (b) moderate (26–40), and (c) high (41–55).

### 2.8. Statistical Analysis

Data will be analyzed by using the Statistical Package in R software. The significance level (*p* < 0.05) will be considered in all statistical tests. All essential R commands will be provided and clearly described to conduct and report the analyses:


**Phase 1**
Aim 1

The systematic review answers a defined research question by collecting and summarizing all empirical evidence that fits the pre-specified eligibility criteria. The meta-analysis uses statistical methods to summarize the results. However, it is subject to the data availability after the systematic review. This study will use tabulation and grouping techniques to extract the data and narrative synthesis. A template and specific guidelines will be used to extract key methodological detail for each paper.


**Phase 2**
Aim 1

A descriptive analysis of the sociodemographic characteristics of the participants will be performed using two R packages; the ‘**gtsummary**’ and ‘**epiR**’ packages. All variables will be assessed for the normality distribution. The median and interquartile range (IQR) will be reported for numerical variables with a skew distribution. Meanwhile, the frequency of observation and its percentage will be reported for categorical variables.

Aim 2

We will perform a three-level general linear mixed model with registered nurses nested within departments and hospitals to model the variability explained by individual-level and group-level variables taking the correlated data structure into account. In this study, the ‘**lme4**’ package in Rstudio IDE will be used [24].

Aim 3

The standard procedure for analyzing causal mechanisms in applied research is called mediation analysis, where a set of linear regression models are fitted and then the estimates of “mediation effects” are computed from the fitted models [25]. In this study, the R package ‘**mediation**’ will be used to analyze mediation effects and ‘**processR**’ for moderation. 

Aim 4

Multiple comparisons will be applied to compare CT and CDM among registered nurses between government, private, and educational hospitals. The **‘multcomp’** package in Rstudio IDE will be used in this study to analyze the comparisons across the hospital settings. The ‘**multcomp**’ package formulates simultaneous inference procedures in situations that were previously hard to deal with [26].

## 3. Expected Results

The primary outcomes of this study include the prevalence of critical thinking and clinical decision-making skills among registered nurses in Malaysia. This study will also document the level of CT and CDM skills of the study participants and understand the factors that influence CT and CDM skills. Specifically, this study will (a) determine the predictive factors on various individual and group levels, (b) quantify the contribution of critical thinking to clinical decision-making formation, and (c) estimate whether the in-service training affects the direction of the relationship between nurses’ knowledge, critical thinking skills, and clinical decision-making skills.

## 4. Discussion

Critical thinking and clinical decision-making skills are at the core of nursing services, yet both are poorly understood. Research shows that individual and organizational characteristics play a significant role in patient safety and the quality of care in nursing practice, but the relative contributions of these characteristics remain unclear [27]. Limited studies also focus on the relationship between group-level factor and CT and CDM skills. To our knowledge, this is the first study to assess CT and CDM using multilevel modeling. By including individual- and group-level factors in the model, the most significant factors that may have a greater influence on CT and CDM skills can be assessed. The results of this study will provide additional information to policymakers to further improve the quality of nursing services and their improvement.

Previous studies have shown that currently available instruments are not sensitive to measuring skills in nursing practice [28,29]. Therefore, one of the products of this study is a validated instrument to measure the critical thinking and clinical decision-making scale in Malay language. In addition, this study will also translate and validate the nurses’ knowledge instrument to the Malay version. This instrument can assess the score of the nurses’ perceived knowledge of CDM. During validation process, we will be using structural equation modelling (SEM) because SEM is a powerful tool and it provides more accurate confirmatory factor analysis results.

There are several limitations that we have identified. First, this study will be conducted only selected hospitals (based on the regions), which may result in limited generalizability. However, multistage stratified cluster random sampling and adequate sample size will minimize this limitation. Secondly, using a self-reported, online survey may lead to response bias (reduce the accuracy of the data collection). In addition, the respondents, due to social desirability, may answer the items in ways to reflect well on themselves. To minimize the effect of social desirability, the instruments used in this study include complex tasks and negative statements. The tasks and statements will reduce the acquiescent bias and increase the confidence and honesty of the participants in answering the questionnaire.

## 5. Conclusions

This manuscript describes a protocol for conducting research to model critical thinking and clinical decision-making skills among registered nurses. The protocol describes two main components of the research project that is (a) the systematic review and (b) the multilevel modeling. The systematic review will provide information on the levels, predictive variables, and instrumentation used in previous studies and the multilevel modeling framework will generate new findings about the nurse’s critical thinking and clinical decision-making skills. The protocol is useful for researchers who want to study the individual and organizational characteristics that play a significant role in patient safety and quality of care among nurses.

## Figures and Tables

**Figure 1 healthcare-11-01169-f001:**
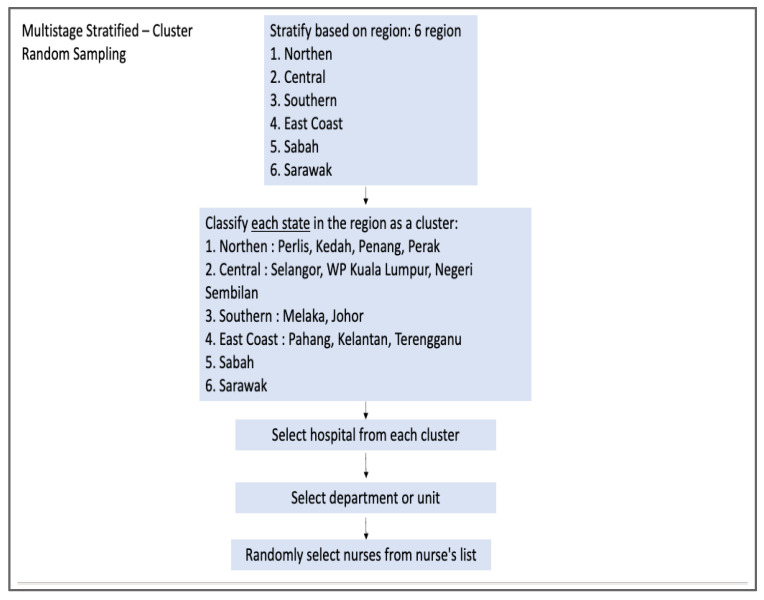
Multistage stratified cluster random sampling for the selection of general hospitals and district hospitals.

**Figure 2 healthcare-11-01169-f002:**
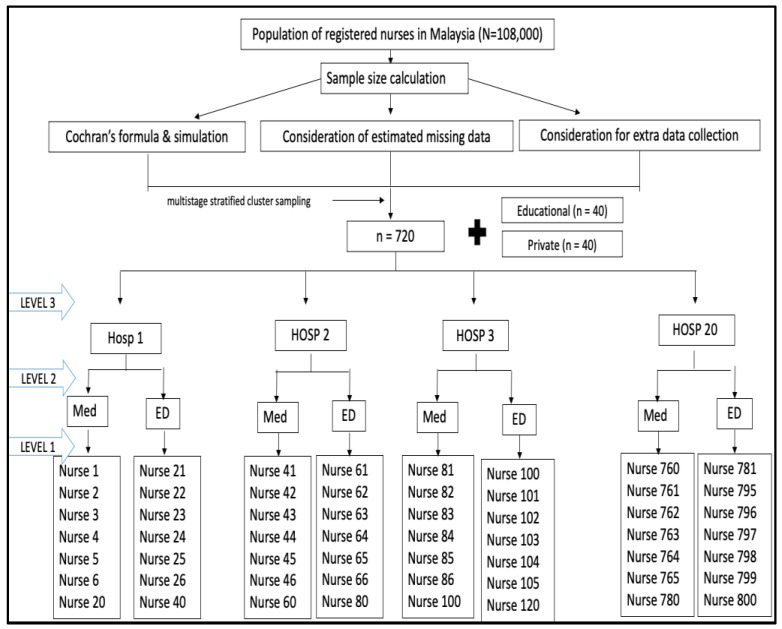
Subject recruitment.

**Figure 3 healthcare-11-01169-f003:**
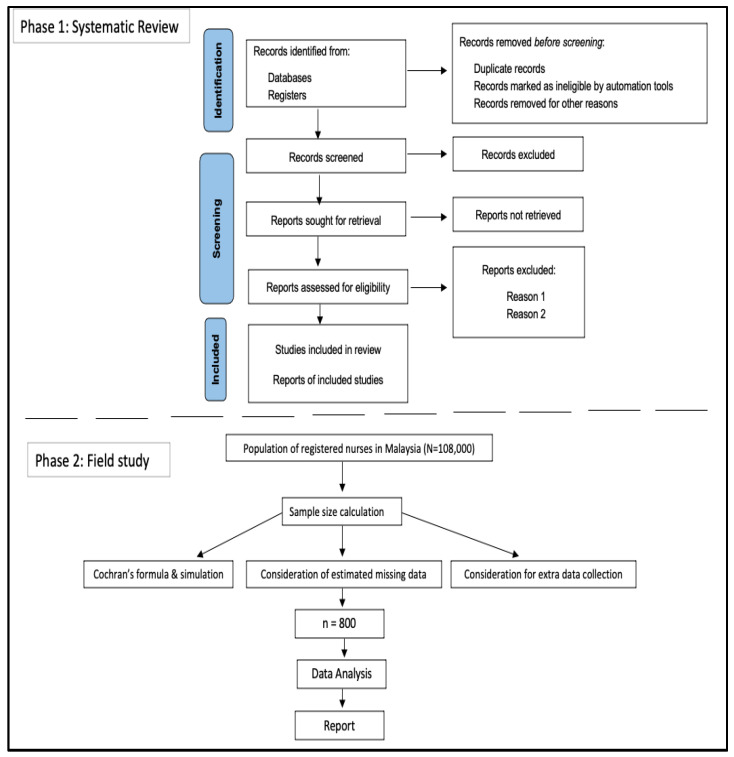
Study flow chart.

**Figure 4 healthcare-11-01169-f004:**
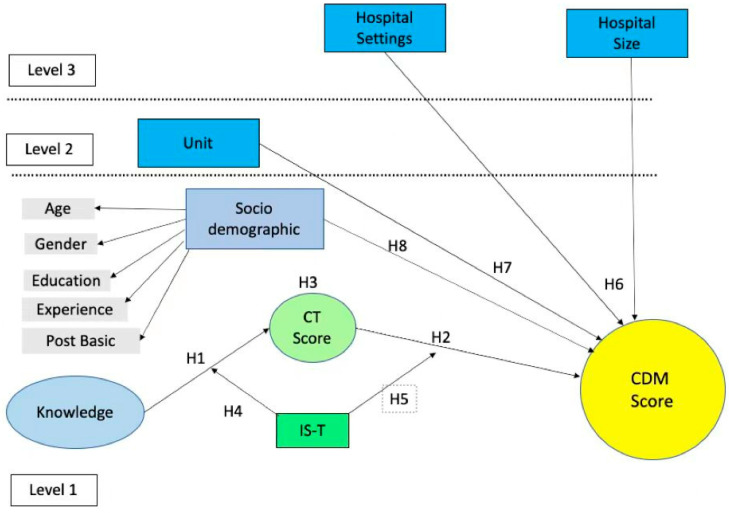
Conceptual framework of multilevel modeling of critical thinking and clinical decision-making skills.

## Data Availability

Not applicable.

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
