# Peer review of "Multilevel Modeling of Individual and Group Level Influences on Critical Thinking and Clinical Decision-Making Skills among Registered Nurses: A Study Protocol"

_healthcare, 2023, doi:10.3390/healthcare11081169_

Round 1

Reviewer 1 Report

- Authors must change the title because now it is confusing. 

-Authors must distinguish a multilevel evaluation framework from a multi-criteria evaluation model. They are different concepts. Now, it is not clear.

- Authors must express numerical itemizes in a correct way. See like an example, line 67.

_ I would like to know the question that the authors made nurses in order to evaluate the method.

- Figure 3 must be completed. Now there are a lot of itemizes incomplete.

- Data analysis is not clear. The authors must explain in detail how they have summarized the information.

- Conclusions does not any relevant results.

- English needs to be reviewed.

Reviewer 2 Report

It is an interesting article. This article presents, justifies, and describes usefully the guidelines to conduct research by using a multilevel modeling framework that wins order to propose a protocol about critical thinking and clinical decision-making skills among nurses. This paper describes the protocol (it is an ongoing project), which aims to examine the prevalence of critical thinking and clinical decision-making and determine the factors that influence the skills at individual and group levels using a multilevel modeling approach. This protocol describes the guidelines to conduct research by using a multilevel modeling framework that will generate findings about the critical thinking and clinical decision-making skills among nurses that are widely debated among researchers. It is a long-debated topic among experts. In nursing practice, critical thinking and clinical decision-making are two essential components; they are laid at every level and daily. Nursing is a very engaged complex of tasks, crucial ones, that involves personnel in clinical decision-making.

Reviewer 3 Report

The manuscript is dealing with a "hot topic" of nursing services: critical thinking and critical decision-making. There are several papers available on these matters. In that sense, the authors need to improve their approach and give a more relevant literature overview. Here are some other comments and suggestions:

1. Literature overview of the previous research findings in the field of Critical thinking and Clinical Decision Making should be added in a separate section "Background research".

2. More references should be added to the following paragraphs:

“Most of the literature focused on CT and CDM levels and factors from the individual level that influence the skills.”

“The minimum sample size was calculated using the Cochran formula with around 73 384 minimum samples.”

“Participants will be recruited using a multistage stratified cluster random sampling.”

“Generally, multilevel modeling begins with the determination of randomly varying outcome parameters which include a variation on the level of the outcome (intercepts) and the determination of within-group relationship that is indicated by the regression coefficients (slope) across groups.”

“The standard procedure for analyzing causal mechanisms in applied research is called mediation analysis, where a set of linear regression models are fitted and then the estimates of “mediation effects” are computed from the fitted models...”

“The nurse’s perceived knowledge instrument will be used to measure the perceived knowledge of nurses regarding CDM skills. This questionnaire consists 145 of 22 items. A validation study will be conducted to provide validated and reliable instruments to be used among local respondents…”

3. The research protocol for the multilevel modeling framework is well-defined but, is there a need for a new modeling framework? In what ways your multilevel modeling framework is superior to others stated in the literature? Or, why there is a need for a new framework? Please explain, and add references.

a) The Conceptual framework paragraph needs some references.
b) The Instrument paragraph needs more work. How have you decided about the items of the questionnaire? Is the questionnaire based on the literature? Please describe, and add references.
c) How do you suppose to handle data and Hypothesis questions? What statistical tests do you intend to use? If you need some inspiration, check this: “A Novel, Modular Robot for Educational Robotics Developed Using Action Research Evaluated on Technology Acceptance Model
d) In the Data analysis paragraph, you describe what you will do with your data but you don’t provide why you choose these specific statistical tests. Add some references too.

3. Your paper needs some results even at an early stage: e.g. systematic review. Otherwise, what is your contribution to the research?

4. Based on Turnitin (turnitin.com) your similarity (plagiarism) is 27%. You have to lower this.

Round 2

Reviewer 1 Report

Authors have not followed reviewer instruction, therefore from my point of view the paper must be reject

Author Response

Dear reviewer 1, we have made another correction to the protocol manuscript. Please see the attachment.

Thank you.

Reviewer 3 Report

Well done! No more comments/suggestions are needed.

Author Response

Dear Reviewer,

Thank you for your feedback.